# A Pharmacometric Model to Predict Chemotherapy-Induced Myelosuppression and Associated Risk Factors in Non-Small Cell Lung Cancer

**DOI:** 10.3390/pharmaceutics14050914

**Published:** 2022-04-22

**Authors:** Kyemyung Park, Yukyung Kim, Mijeong Son, Dongwoo Chae, Kyungsoo Park

**Affiliations:** 1Department of Pharmacology, Yonsei University College of Medicine, Seoul 03722, Korea; pkm304@gmail.com (K.P.); ykkim@yuhan.co.kr (Y.K.); mijeong.son@boryung.co.kr (M.S.); dongy@yuhs.ac (D.C.); 2Brain Korea 21 Plus Project for Medical Science, Yonsei University, Seoul 03722, Korea

**Keywords:** neutropenia, pharmacokinetic and pharmacodynamic modeling, non-small cell lung cancer, chemotherapy, paclitaxel, cisplatin, myelosuppression, transit compartments, K-PD model, R Shiny

## Abstract

Chemotherapy often induces severe neutropenia due to the myelosuppressive effect. While predictive pharmacokinetic (PK)/pharmacodynamic (PD) models of absolute neutrophil count (ANC) after anticancer drug administrations have been developed, their deployments to routine clinics have been limited due to the unavailability of PK data and sparseness of PD (or ANC) data. Here, we sought to develop a model describing temporal changes of ANC in non-small cell lung cancer patients receiving (i) combined chemotherapy of paclitaxel and cisplatin and (ii) granulocyte colony stimulating factor (G-CSF) treatment when needed, under such limited circumstances. Maturation of myelocytes into blood neutrophils was described by transit compartments with negative feedback. The K-PD model was employed for drug effects with drug concentration unavailable and the constant model for G-CSF effects. The fitted model exhibited reasonable goodness of fit and parameter estimates. Covariate analyses revealed that ANC decreased in those without diabetes mellitus and female patients. Using the final model obtained, an R Shiny web-based application was developed, which can visualize predicted ANC profiles and associated risk of severe neutropenia for a new patient. Our model and application can be used as a supportive tool to identify patients at the risk of grade 4 neutropenia early and suggest dose reduction.

## 1. Introduction

Chemotherapy is a major pharmacotherapeutic modality for treating cancer. One of the major side effects of chemotherapy is severe (grade 4) neutropenia due to the myelosuppressive effect of cytotoxic drugs [1]. Once grade 4 neutropenia occurs, physicians reduce the next round dose of anticancer drugs by prespecified proportion, which may result in suboptimal anticancer effects. Therefore, predictive models of neutrophil count dynamics after drug administration can help identify vulnerable patients in need of close monitoring and prophylactic treatments and determine accurate dose reduction that prevents the risk of severe neutropenia and continue to receive anticancer treatments. Several semi-mechanistic pharmacokinetic and pharmacodynamic (PK/PD) models for the effects of myelosuppression by anticancer drugs and recovery by granulocyte colony stimulating factor (G-CSF) have been developed [2,3,4,5].

However, the deployment of such models in daily clinics has been limited [6]. Existing relevant PK/PD models were usually developed using rich data with drug concentrations intensively sampled and absolute neutrophil count (ANC) frequently measured. However, in the usual clinical setting with sparse data, where drug concentrations are often unavailable and ANC tends to be measured less frequently, existing models based on rich data are unsuitable. A promising way to overcome the lack of drug concentration data has been to use kinetic-pharmacodynamic (K-PD) modeling on top of the existing PD model [5,7].

Lung cancer is one of the most frequent causes of cancer-related death worldwide. Since lung cancer is usually diagnosed at the late stages (stages III or IV), pharmacologic treatments are often the initial treatment of choice. Although new innovative pharmacological agents for targeted therapies or immunotherapies have appeared, cytotoxic chemotherapies are still used in a significant portion of lung cancer patients. Therefore, the modeling framework for assessing and monitoring lung cancer patients for severe neutropenia in routine clinics is of great importance.

In this study, we aimed to develop a K-PD model of ANC dynamics and identify influencing factors that can be applied to the routine clinical setting of lung cancer chemotherapy. In doing so, data retrospectively collected from electrical medical records (EMRs) for non-small cell lung cancer (NSCLC) patients who underwent a combined (cisplatin and paclitaxel) chemotherapy regimen were used. Then, using the ANC model developed, we proposed a web application to predict the risk of severe neutropenia for individual patients based on sparsely sampled ANC observations.

## 2. Methods

### 2.1. Data

The data were retrospectively collected from EMRs of the patients who received lung cancer treatment at Severance Hospital, Yonsei University, Seoul, Korea, between January 2009 and December 2013.

A total of 6058 lung cancer patients were screened. Among these patients, first-line combination treatment of paclitaxel and cisplatin was given to 327 patients and, finally, 173 were included in the analysis. They met the following selection criteria: the inclusion criteria being between 18 and 85 years old and cancer stage of IIIB or IV and the exclusion criteria being the history of the primary tumor removed by surgery, concurrent chemoradiation, or no evaluation for the treatment effect.

The treatment was conducted in up to six cycles of 28 days each, with paclitaxel infused for 3 h at the rate of 175 mg/m^2^ (with m^2^ denoting unit body surface area) on day 1 and cisplatin infused for 3 h at the rate of 75 mg/m^2^ on day 2 of each cycle [1], where the dosing was reduced by 25% if severe neutropenia (grade 4) occurred or stopped when disease progression or condition deterioration was observed. The response was evaluated every three cycles and whenever disease progression was doubtful. The dependent variable analyzed was ANC obtained as ANC = white blood cell count × the percentage of neutrophils ÷ 100. Potentially influential covariates collected included demographic factors, disease history, laboratory markers of hepatic and renal function, and cancer-related indices. In detail, demographic factors included height, body weight (WT), body surface area (BSA), age when diagnosed, sex, and smoking history; disease history included hypertension (HTN), diabetes mellitus (DM), and pulmonary tuberculosis (TB); hepatic function included aspartate aminotransferase (AST) and alanine transaminase (ALT); renal function included serum creatinine, estimated creatinine clearance (CLcr), and estimated glomerular filtration rate (eGFR) where CLcr was calculated using Cockroft–Gault formula [8] and eGFR was estimated using a modification of diet in renal disease formula [9,10]; and cancer-related indices included disease stage, baseline Eastern Cooperative Oncology Group (ECOG) performance status, baseline tumor size, histology, overall response, and performed chemotherapy cycles. G-CSF treatment information was also incorporated into the model, assuming that the effect of filgrastim (Grasin prefilled injection^®^, Leukokine injection^®^, and Leucostim injection^®^) and lenograstim (Neutrogin injection^®^), G-CSF analog formulations used in this work, were the same as G-CSF.

In most cases, ANC data comprised sparse samples collected only twice each cycle, with additional samples taken for the G-CSF treatment. WT, BSA, and hepatic and renal function were allowed to change with each cycle. Missing covariate information was replaced by the information at the previous sampling time or the mean value of two neighboring (previous and next) sampling times.

### 2.2. Model Development

#### 2.2.1. Semi-Mechanistic Myelosuppression Model

Following the work by Friberg et al. [2], the pharmacodynamics of neutrophil count was described using a semi-mechanistic model as depicted in Figure 1. The model comprised two single compartments representing proliferative cells [*Prol*] and circulating observed blood cells [*Circ*], interconnected with a series of transit compartments [*Transit*] and associated rate constant (*Ktr*) representing a time delay needed for proliferative cells to mature into circulating blood cells. A proliferation rate constant (*Kprol*) represented the multiplication rate of proliferative cells, which is controlled by negative feedback from the circulating cells (*Circ*_0_/*Circ*)*^γ^*, with *γ* reflecting self-replication rate increases when circulating cell levels decrease compared with the baseline value (*Circ*_0_). Then, *Kprol* was assumed to be inhibited by drug effect (*E_d_*) and *Kprol* and *Ktr* stimulated by G-CSF effect (*E_G_*). The model equations are summarized in Equations (1)–(7).
(1)$$\frac{dProl}{dt}=Kprol\cdot Prol\cdot E_{d}\cdot {\left(\frac{Circ_{0}}{Circ}\right)}^{\gamma }-Ktr\cdot Prol$$
(2)$$\frac{dTransit1}{dt}=Ktr\cdot Prol-Ktr\cdot Transit1$$
(3)$$\frac{dTransit2}{dt}=Ktr\cdot Transit1-Ktr\cdot Transit2$$
(4)$$\frac{dTransit3}{dt}=Ktr\cdot Transit2-Ktr\cdot Transit3$$
(5)$$\frac{dCirc}{dt}=Ktr\cdot Transit3-Kcirc\cdot Circ$$
(6)$$Ktr=Ktr_{0}\cdot E_{G}$$
(7)$$Kprol=Kprol_{0}\cdot E_{G}$$
where *Ktr*_0_ and *Kprol*_0_ are pre-treatment values of *Ktr* and *Kprol* evaluated at the baseline (i.e., *E_G_* = 1), respectively.

The basic assumption for the model is that, at a steady state without drug treatments, *dProl*/*dt* = 0, and therefore, *Kprol*_0_ = *Ktr*_0_. Due to numerical difficulty, it was assumed that *Kcirc* = *Ktr*_0_. To improve interpretability, instead of *Ktr*_0_, the mean transit time *MTT* = (*n* + 1)/*Ktr*_0_ (*n*: number of transit compartments) was estimated.

#### 2.2.2. Kinetic-Pharmacodynamic (K-PD) Drug Model

As blood concentration data were not available for anticancer drugs and G-CSF, a K-PD model was used [5,7], which described drug kinetics using a virtual one-compartment model with bolus input as follows:(8)$$\frac{dA_{d}}{dt}=-KDE_{d}\cdot A_{d}$$
(9)$$VIR_{d}=KDE_{d}\cdot A_{d}$$
where *A_d_* represents the amount of drug in the hypothetical compartment, *KDE_d_* the elimination rate constant from the virtual compartment, and *VIR_d_* the virtual infusion rate of drugs distributed into PD sites.

#### 2.2.3. Drug Effect of Combination Chemotherapy

With the K-PD model defined in Equations (8) and (9), the combination drug effect was described by either the log-linear model (Equation (10)) or the response surface model developed by Minto et al. (Equation (11)) [11].
(10)$$E_{d}=e^{-Scale1\cdot VIR_{P}-Scale2\cdot VIR_{C}}$$
(11)$$E_{d}=1-\frac{{\left(\frac{\frac{VIR_{P}}{IR_{50,P}}+\frac{VIR_{C}}{IR_{50,C}}}{U_{50}}\right)}^{p}}{1+{\left(\frac{\frac{VIR_{P}}{IR_{50,P}}+\frac{VIR_{C}}{IR_{50,C}}}{U_{50}}\right)}^{p}}$$
where Scale represents the slope, *IR*_50,*P*_ and *IR*_50,*C*_ represent *VIR* of each drug producing a half maximal effect, *U*_50_ is the number of units corresponding to half maximal effect at the given drug combination ratio, and *p* is the steepness of drug concentration–response relation. Using the response surface model, drug interaction type (additive, synergistic, or antagonistic) and severity can be assessed.

#### 2.2.4. G-CSF Effect

Similar to the drug effect, the G-CSF effect, which was assumed to affect myelocytes continuously, was analyzed using the K-PD model (Equations (12) and (13)), followed by the linear (Equation (14)), the ordinary Emax (Equation (15)) and the constant model (Equations (16) and (17)). The model for G-CSF effect was then incorporated into the *E_G_* of *Ktr* (Equation (6)) and *K**prol* (Equation (7)) because G-CSF shortens the *MTT* and increases the mitotic activity [5,12].
(12)$$\frac{dA_{G}}{dt}=-KDE_{G}\cdot A_{G}$$
(13)$$VIR_{G}=KDE_{G}\cdot A_{G}$$
(14)$$E_{G}=1+Scale3\cdot VIR_{G}\;\;:\;Linear\;model$$
(15)$$E_{G}=1+\frac{VIR_{G}}{VIR_{G}+IR_{50,G}}\;\;\;:\;\;Emax\;model$$
(16)$$E_{G}=1+\theta_{Ktr}\cdot GCSF\;\;\;:\;\;\;\;Constant\;model\;for\;Ktr$$
(17)$$E_{G}=1+\theta_{Kprol}\cdot GCSF\;\;:\;\;\;\;Constant\;model\;for\;\;Kprol$$

For the constant models, we followed the work by Ramon-Lopez et al. [5], where $\theta_{Ktr}$ and $\theta_{Kprol}$ are the relative contribution of the G-CSF effect to *Ktr* and *Kprol*, respectively, and *GCSF* is an indicator variable denoting the value of 1 for the G-CSF treatment and 0 otherwise.

However, due to a high correlation between $\theta_{Ktr}$ and $\theta_{Kprol}$ (see the Results section for detail), we tried the following model also:(18)$$E_{G}=\left(1+\theta_{Ktr}\cdot GCSF\right)\cdot \left(1+\theta_{Kprol}\cdot GCSF\right)\;\;:\;\;Constant\;model\;for\;\;Kprol$$

This splits the effect of G-CSF on *Kprol* into that shared with *Ktr* and that solely on *Kprol*, effectively removing the correlation.

### 2.3. Covariate Modeling

The covariate effects on the structural parameters were tested for BSA, ALT, CLcr, sex, age, HTN, DM, TB, smoking history, and ECOG performance status, using linear, exponential, and power models. In doing so, assuming that ΔOBJ (the difference in NONMEM objective function values) between the models with and without a covariate was approximately χ^2^-distributed, the stepwise covariate modeling (SCM) approach was used in such a way that at one degree of freedom a covariate yielding ΔOBJ ≥ 6.635 (or *p*-value ≤ 0.01) for the forward inclusion and ΔOBJ ≥ 10.828 (or *p*-value ≤ 0.001) for the backward deletion was included in the model.

### 2.4. Statistical Model

For statistical model building, interindividual variability (IIV) was described using an exponential error model and residual unexplained variability using a combined error model, assuming that they are normally distributed with mean zero and variance ω^2^ and σ^2^, respectively. Due to numerical issues, IIV was included in only part of the model parameters; in the response surface model, for example, it was included only in *Circ*_0_, *MTT*, *IR*_50,_ and KDE.

### 2.5. Model Evaluation

The model diagnosis was made based on the Akaike information criterion (AIC), followed by goodness-of-fit (GOF) plots, precisions of model parameter estimates, and finally visual predictive check (VPC) plots using 1000 datasets simulated from the final model.

### 2.6. A Web Application for a New Patient’s ANC Prediction

Based on the final model developed, we developed an R Shiny web application for a new patient, which can visualize (i) the typical trend of ANC time-profile predicted at the baseline when no ANC observation is available and (ii) the individual trend of ANC time-profile predicted using individual estimates of the model parameters obtained from ANC observations available after the chemotherapy begins. Here, the individual parameter estimates were obtained as the maximum a posteriori (MAP) estimate using the Monte Carlo Markov Change sampler in the R *adaptMCMC* package. The individual ANC time profile was then used to estimate the probability of lethal neutropenia [13].

### 2.7. Software

All analyses were conducted using NONMEM version 7.4.4 (ICON Development Solutions, Ellicott City, MD, USA), and the first-order conditional estimation with interaction (FOCE inter) method was used for model building. PsN version 5.2.6 was used for SCM [14,15]. Model diagnostics of GOF and VPC plots were produced using PsN and R program version 3.6.2.

## 3. Results

### 3.1. Data

Here, 828 cycles of combination chemotherapy were given to the patients, and approximately two ANC samples were taken before and after each treatment cycle, resulting in 1686 ANC observations collected in total, with the mean baseline of 5.5 × 10^9^ cells/L. With G-CSF treatment given whenever grade 4 neutropenia was reported, 37 patients received 63 G-CSF treatments, all together. The detail of the subjects’ characteristics is shown in Table 1.

### 3.2. Basic Model for Drug Effect

The data of 136 (=173 − 37) patients who were not treated with G-CSF were used for basic model building. Using the semi-mechanistic model, the analysis was conducted with 1 to 3 transit compartments. For the combined effect of the drugs, the response surface model assumed that *IR*_50_ or *KDE* of paclitaxel and cisplatin were the same. The log-linear or additive model was not estimable probably because both paclitaxel and cisplatin inhibit ANC production in the same (i.e., proliferation) compartment.

The model assuming that *IR*_50_ was shared between paclitaxel and cisplatin (*IR*_50,*p*_
*= IR*_50,*c*_) showed more reasonable estimates and better precision of parameters than the one sharing *KDE*, probably because the two drugs were given almost at the same time and their maximum effects could not be distinguished, and therefore the model with the shared *IR*_50_ produced more reliable results. With the assumption of *IR*_50,*p*_ = *IR*_50,*c*_, the model with three transit compartments yielded the lowest AIC, consistent with Friberg et al. [2].

### 3.3. Basic Model for Drug and G-CSF Effects

The model for drug effect, selected using data from 136 patients, was used as a template to build the model for the entire 173 patients, including 37 receiving G-CSF treatment.

K-PD models (Equations (12) and (13)) resulted in poorer estimations than the constant model (Equations (16) and (17)). Moreover, the estimate of $KDE_{G}$ showed a high correlation with Scale3 and $IR_{50,G}$ (Equations (14) and (15)), respectively.

For the constant model, because the G-CSF effect lasted only 1.5 days after the administration [4], it resulted in unreliable estimates of $\theta_{Ktr}$ and $\theta_{Kprol}$ with a high correlation (0.97) between them (data not shown).Therefore, the model was modified as in Equation (18), which splits the effect of G-CSF on *Kprol* into that shared with *Ktr* and that solely on *Kprol*, effectively removing the correlation. Accordingly, the constant models (Equations (16) and (18)) were finally selected.

### 3.4. Covariate Modeling

The selected covariates were *DM* and *sex* for *IR*_50_ and they were formulated as:(19)$$IR_{50}=\theta_{TV}\cdot \left(1+\theta_{sex}\cdot SEX\right)\cdot \left(1+\theta_{DM}\cdot DM\right)$$
where *SEX* is 1 for female and 0 for male, and *DM* is 1 for patients with *DM* and 0 for those without *DM*. Both affected *IR*_50_, but in opposite directions. With the estimates of $\theta_{sex}$ and $\theta_{DM}$ being obtained as −0.334 and 0.485, respectively, *IR*_50_ for female decreased by 0.334 fold or 33.4% as compared to men, and *IR*_50_ for DM patients increased by 0.485 fold or 48.5% as compared to non-DM patients, indicating that female patients are more prone to neutropenia and DM patients are less prone to neutropenia. The estimates of the final model parameters are shown in Table 2.

### 3.5. Model Evaluation

Parameter estimates of the final model were evaluated with a bootstrap analysis by generating 500 replicates of data. The resulting bootstrap estimates and RSE values are reported in Table 2, which agreed well with those obtained by NONNEM analysis, confirming the identifiability and reproducibility of the final model parameters.

The goodness-of-fit plots for the final model displayed in Figure 2 show that observations are evenly distributed around the line of identity or zero residual line except for a few points, indicating the model describes the data well. Regarding the points out of the identity or zero residual line, it was conjectured that they were related to G-CSF treatment, where ANC after G-CSF treatment was predicted to be slightly higher than the observed value. The VPC plot illustrated that the model with estimated parameters reliably predicts the typical trend and variation of ANC time courses (Figure 3).

### 3.6. Sub-Group Analyses

We investigated the appropriateness of the selected covariates SEX and DM by comparing observed and predicted ANC profiles for covariate subgroups as plotted in Figure 4a,b. The figure shows that predicted profiles well represent observed profiles in both covariate subgroups, revealing that female and non-DM patients were more susceptible to chemotherapy-induced neutropenia than male and DM patients, respectively. Age and body size (body weight and BSA) showed significant associations with DM status, thereby raising the possibility of being confounding factors for DM (Table 3).

### 3.7. A Web Application for a New Patient’s ANC Prediction

The R Shiny package developed (https://pkm304.shinyapps.io/NSCLC_neut/ accessed on 15 March 2022) is illustrated in Figure 5 for the individual ANC time profile of a new patient predicted from the final model. Users can enter patient demographic information such as BSA, baseline ANC, DM status, and sex by manipulating slide bars and radio buttons (Figure 5a). Then, a dosing schedule can be provided by clicking the “add” buttons for paclitaxel/cisplatin and/or G-CSF and entering “days” and “dose per BSA” (Figure 5b). With the additional input of observed ANCs of the given patient, individual parameter estimates and their posterior distributions are generated (Figure 5c). Based on this, ANC profiles with posterior variabilities can be simulated and plotted (Figure 5d). Finally, the risk of grade 4 neutropenia can be assessed based on the predicted trajectories (Figure 5e).

## 4. Discussion

In this study, we developed a K-PD model to predict ANC dynamics during chemotherapy in NSCLC patients, which can be used to assess the risk of lethal neutropenia. For this, we applied existing models [2,5] developed with rich data to routine clinical settings characterized by sparse patient-derived data with infrequent ANC measurements and lacking in drug concentration measurements. We showed that the derived model could be well-calibrated to the sparse dataset retrospectively collected from the EMR system. The K-PD models have been successfully used for PD analyses in drug development when drug concentration data were not available [7,16]. Furthermore, we developed a web application to visualize predicted ANC time courses and estimate the risk probability of severe neutropenia during chemotherapy to accelerate the bedside utilization of the developed model.

For the selected covariates of sex and DM in our work, the risk for chemotherapy-induced neutropenia was higher in women than in men (Figure 4a), which is consistent with the result reported previously [17]. On the other hand, DM patients were found less susceptible to neutropenia as compared to non-DM patients (Figure 4b), while little evidence was reported about such protective effect of DM [18] and hematopoietic stem cell mobilization was reported to be impaired in DM patients [19,20]. Our result, inconsistent with previous reports on the effect of DM, might be explained in two ways. One possible explanation would be associated with the dose of anti-cancer drugs used in our study, which was given based on BSA, not WT. In this regard, it was found that the WT of DM patients was significantly higher than non-DM patients (Table 3). As BSA-based dose may tend to be lower than what is required by WT-based dose, this finding might indicate that DM patients would have received a lower dose per Kg than non-DM patients, thereby being likely to show low susceptibility to drug-induced myelosuppression than non-DM patients. Another possibility would be that while previous studies were based on logistic regression analysis of the binary outcome of grade 4 neutropenia or not, our analysis was based on semi-mechanistic modeling of continuous ANC values with time. However, as the protective effect of DM on chemotherapy-induced neutropenia found in our work lacks a physiological basis, more studies, including external validation, would be needed to confirm this finding.

Clinically, the proposed model and the web application can be used in real-time at the bedside to assess the risk of lethal neutropenia in individual patients and suggest close monitoring or prophylactic G-CSF administration for patients with high risks. Moreover, for those who developed grade 4 neutropenia after the previous cycle, dose reduction of the next cycle administration minimally affecting the anticancer effect can be suggested based on simulations using the R Shiny web application (Figure 5).

There are several limitations to this study. First, the data were collected retrospectively from a single center. Additional data would be needed for further validation. Second, the chemotherapy regimens in NSCLC have been updated tremendously along with the rapid development of molecular targeted therapies and immunotherapies [21]. Although the cisplatin/paclitaxel (or other) combination chemotherapy is still indicated for patients with advanced cancers, major preferred therapies include immunochemotherapy combining immune checkpoint inhibitors and platinum-based drugs [21,22]. Therefore, there is an urgent need for model development in those recently developed therapies. Third, the sparse data forced us to rely on trial-and-error processes for choosing the most suitable functional forms of *E**_d_*, *E_G_*, and IIVs. In particular, estimations of IIVs for *E_G_*-related parameters showed poor precisions and therefore were not included (data not shown).

In conclusion, our model successfully described the time course of ANC change caused by paclitaxel/cisplatin-induced myelosuppression and anti-myelosuppression effect of co-administered G-CSF. As illustrated by the R-Shiny web application, our model can be used as a supportive tool to identify patients at risk of grade 4 neutropenia and suggest paclitaxel/cisplatin dose adjustment to prevent the occurrence of lethal neutropenia.

## Figures and Tables

**Figure 1 pharmaceutics-14-00914-f001:**
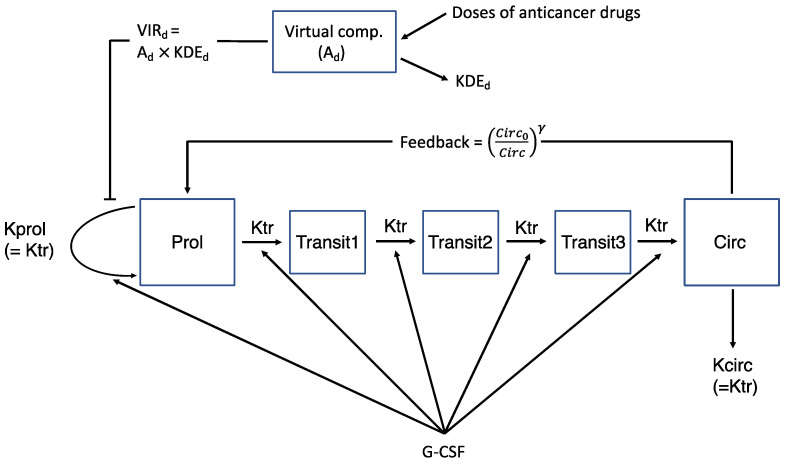
Model schematic.

**Figure 2 pharmaceutics-14-00914-f002:**
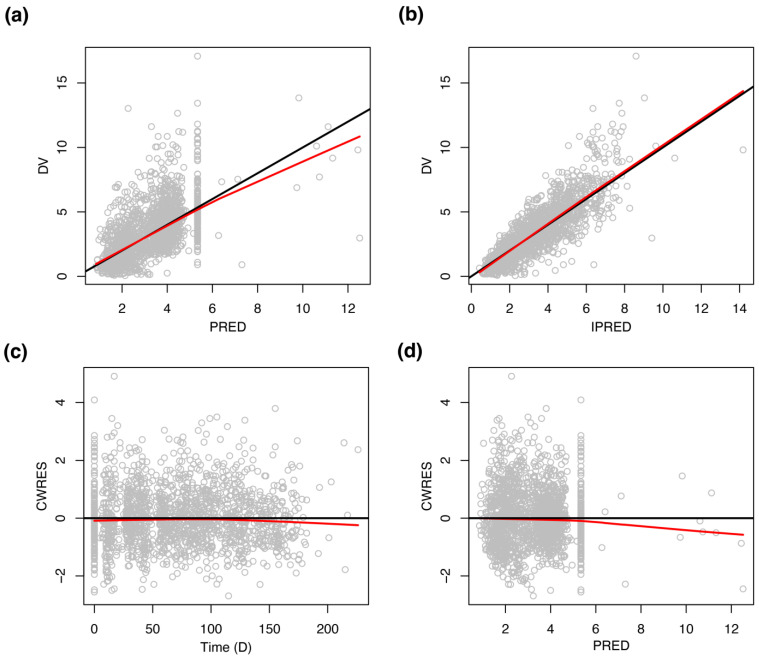
Goodness-of-fit plots of the final model. (**a**) Scatterplot showing DV (observations) vs. typical predictions (PRED). (**b**) Scatterplot showing DV (observations) vs. individual predictions (PRED). (**c**) Scatterplot showing conditional weighted residuals (CWRES) vs. time. (**d**) Scatterplot showing conditional weighted residuals (CWRES) vs. PRED. Black lines represent identity lines for (**a**,**b**) and zero residual lines for (**c**,**d**), and red lines represent smoother lines.

**Figure 3 pharmaceutics-14-00914-f003:**
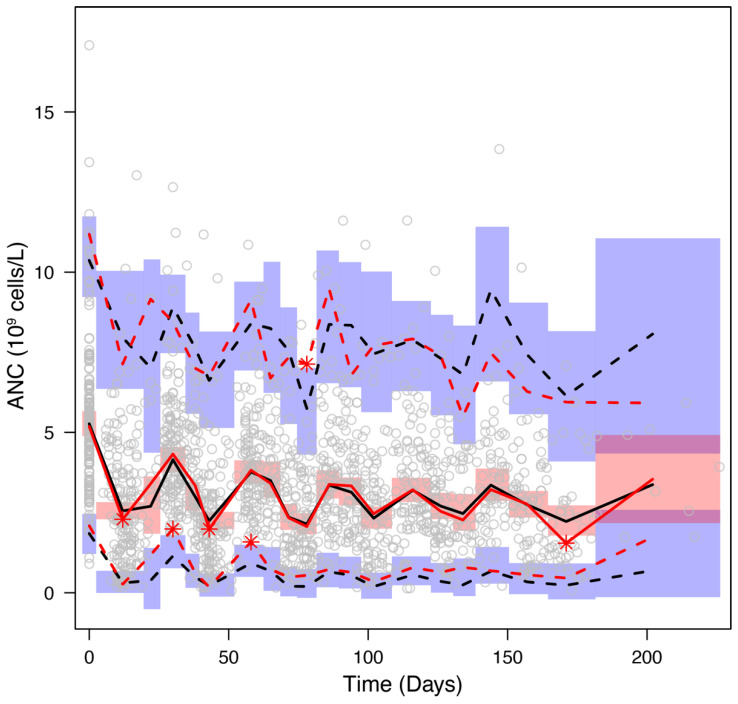
Visual predictive check plot. Open circles represent ANC observations. Red solid and dashed lines represent the median and 5%/95% observed data values, respectively. Black solid and dashed lines represent the median and 5%/95% simulated data values, respectively. Red areas indicate the 95% confidence intervals on the simulated median. Blue areas indicate the 95% confidence intervals on the simulated 5% and 95% values.

**Figure 4 pharmaceutics-14-00914-f004:**
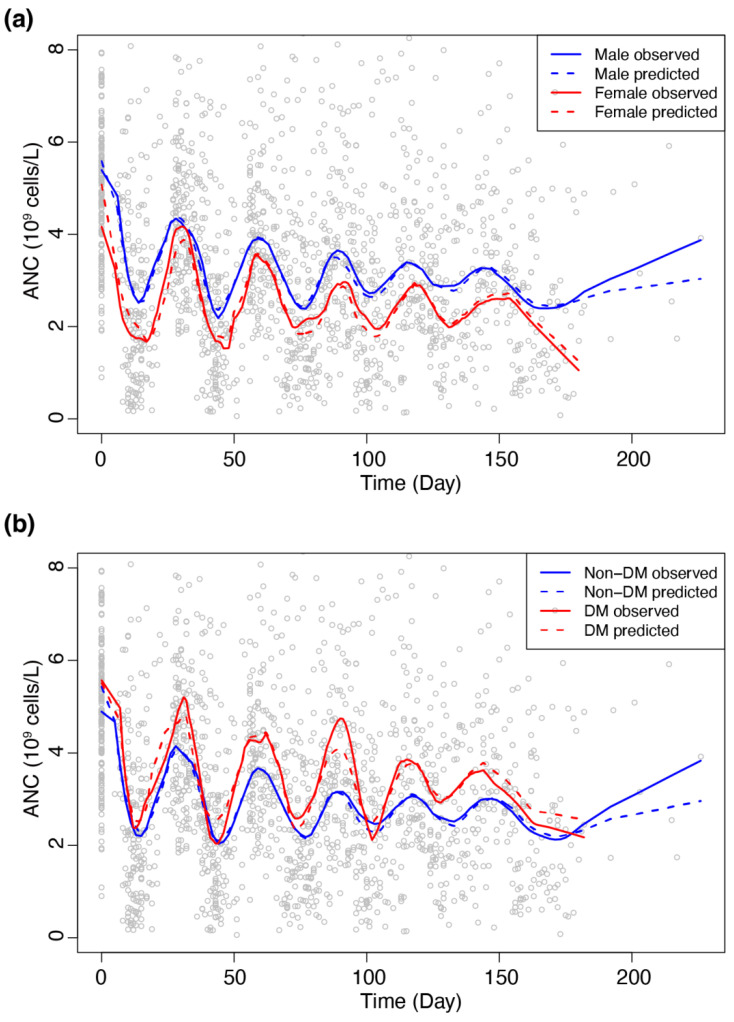
Averaged trajectories stratified with the selected covariates. Open circles ANC depict observations. Solid lines represent averaged observation. Dashed lines represent averaged predictions. (**a**) Blue: male, red: female. (**b**) Blue: DM, red: non-DM.

**Figure 5 pharmaceutics-14-00914-f005:**
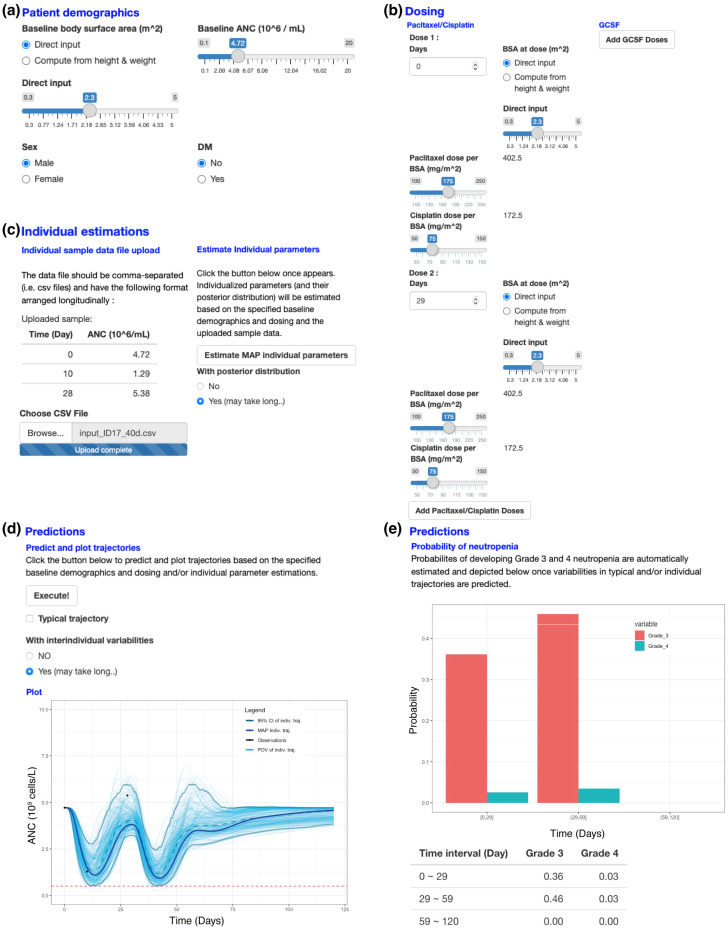
An illustrative example of the Shiny application. (**a**) Patient demographics. (**b**) Dosing schedule specification. (**c**) User input of observed ANC and individual estimation of parameters with their posterior distributions. (**d**) Simulation based on the estimated parameters and posterior distributions. (**e**) Assessment of the probability of grade 4 neutropenia.

**Table 1 pharmaceutics-14-00914-t001:** Patient demographic characteristics.

(a) Continuous Variables		
Variable	Mean (±SD)	Median (Min, Max)
Baseline ANC (10^6^ cells/L)	5521 (±2328)	5183 (901, 17,080)
White blood cell count (10^9^ cells/L)	8.25 (±2.58)	7.93 (2.14, 20.63)
Height (cm)	164.3 (±8.6)	165.0 (140.0, 186.0)
Body weight (kg)	63.5 (±11.3)	63.1 (38.0, 107.0)
Body surface area (m^2^)	1.70 (±0.18)	1.70 (1.24, 2.30)
ALT (IU/L)	21.0 (±16.2)	16 (6, 126)
AST (IU/L)	21.4 (±12.8)	18.0 (8.0, 112.0)
Creatinine (mg/dL)	0.86 (±0.19)	0.83 (0.43, 1.82)
Total bilirubin	0.51 (±0.21)	0.50 (0.10, 1.40)
CLcr (mL/min)	81.0 (±23.6)	75.7 (36.8, 153.6)
eGFR (mL/min/1.73 m^2^)	86.2 (±13.3)	90.0 (55.0, 130.6)
Age (years)	61.0 (±9.2)	62.0 (38.0, 82.0)
Primary tumor size (cm) (*n* = 145)	4.7 (±2.3)	4.3 (0.9, 17.0)
**(b) Categorical Variables**		
**Variable**	***n* (%)**	
Sex (male/female)	127 (73.4)/46 (26.6)	
Hypertension (no/yes)	88 (50.9)/85 (49.1)	
Diabetes mellitus (no/yes)	141 (81.5)/32 (18.5)	
Tuberculosis (no/yes)	152 (87.9)/21 (12.1)	
ECOG (0/1)	149 (86.1)/24 (13.9)	
Smoker (Non/Current/Ex)	51 (29.5)/72 (41.6)/50 (28.9)	
Stage (IIIB/IV)	43 (24.9)/130 (75.1)	
Histology (Ad/Sq/Un)	119 (68.8)/50 (28.9)/4 (2.3)	
Overall response (PR/SD/PD)	45 (26.0)/55 (31.8)/73 (42.2)	
Treated cycles (2/3/4/5/6)	8 (4.6)/37 (21.4)/27 (15.6)/12 (6.9)/89 (51.5)	
G-CSF treatment (no/yes)	136 (78.6)/37 (21.4)	

Abbreviations: ANC, absolute neutrophil count; ALT, alanine transaminase level; AST, aspartate aminotransferase level; CLcr, estimated creatinine clearance; eGFR, estimated glomerular filtration rate; Non, non-smoker; Ex, ex-smoker; Ad, adenocarcinoma; Sq, squamous cell cancer; Un, unspecified; PR, partial response; SD, stable disease; PD, progressive disease.

**Table 2 pharmaceutics-14-00914-t002:** Parameter estimates of the final model.

Parameters	Estimate	RSE%	Bootstrap Median	95% CI	Bootstrap RSE%
*Circ*_0_ (10^9^ cells/L)	5.34	2.8	5.34	5.10–5.60	2.8
*MTT* (D)	4.64	3.3	4.69	4.44–5.08	4.2
*γ*	0.188	6.4	0.193	0.173–0.214	6.6
*IR*_50_ (mg/D)	$\theta_{TV}$: 88.9	7.8	86.6	74.0–104.3	10.9
	$\theta_{sex}$: −0.334	16.4	−0.323	−0.433 to −0.222	19.9
	$\theta_{DM}$: 0.485	31.5	0.483	0.222–0.770	35.2
*KDE_p_* (/D)	0.0326	13.2	0.0343	0.0240–0.0497	22.6
*KDE_c_* (/D)	0.188	24.5	0.176	0.122–0.353	40.9
$\theta_{Ktr}$	3.50	8.7	3.48	2.26–4.69	19.8
$\theta_{Prol}$	0.217	8.5	0.226	0.163–0.291	16.2
IIV of *Circ*_0_ (CV%)	23.0	11.4	23.1	20.5–25.8	6.8
IIV of *MTT* (CV%)	16.2	16.3	15.7	12.3–18.7	13.6
IIV of *IR*_50_ (CV%)	23.3	24.5	24.0	12.4–34.3	28.2
IIV of *KDE_p_* (CV%)	95.6	13.8	88.7	58.4–110.9	19.1
IIV of *KDE_c_* (CV%)	92.0	24.1	93.1	70.5–113.5	14.8
**Residual Variability**	**Estimate**	**RSE%**	**Bootstrap Median**	**95% CI**	
σ_prop_ (CV(%))	30.0	3.9	29.3	26.2–32.8	6.6
σ_additive_ (10^9^ cells/L)	0.527	8.5	0.535	0.384–0.663	15.5

The covariate model for $IR_{50}$ is $IR_{50}=\theta_{TV}\cdot \left(1+\theta_{sex}\cdot SEX\right)\cdot \left(1+\theta_{DM}\cdot DM\right)$, where *SEX* is 1 for female and 0 for male, and *DM* is 1 for patients with *DM* and 0 for those without *DM*. Abbreviations: RSE, relative standard error; IIV, interindividual variability; CV, coefficient of variation; TV, typical value; CI, confidence interval.

**Table 3 pharmaceutics-14-00914-t003:** Covariate associations with DM status.

	Non-DM Patients (*n* = 141)	DM Patients (*n* = 32)	*p*-Value
Continuous factors			
Age (years)	60.2 (±9.5)	64.5 (±6.7)	0.004
Height (cm)	164.1 (±8.8)	165.3 (±7.7)	0.418
Body weight (kg)	62.5 (±11.1)	68.1 (±11.2)	0.011
Body surface area (m^2^)	1.68 (±0.18)	1.76 (±0.16)	0.016
ALT (IU/L)	20.1 (±14.5)	25.0 (±22.1)	0.133
AST (IU/L)	21.1 (±11.2)	22.7 (±18.3)	0.726
Creatinine (mg/dL)	0.84 (±0.18)	0.93 (±0.23)	0.036
Total bilirubin	0.50 (±0.20)	0.52 (±0.22)	0.872
CLcr (mL/min)	81.8 (±23.9)	77.7 (±22.3)	0.513
eGFR (mL/min/1.73 m^2^)	87.0 (±13.6)	82.7 (±11.3)	0.056
Categorical factors			
Female sex	40 (28.4%)	6 (18.8%)	0.375
Hypertension	64 (45.4%)	21 (65.6%)	0.050
Tuberculosis	15 (10.6%)	6 (18.8%)	0.231
ECOG			>0.999
0	121 (85.8%)	28 (87.5%)	
1	20 (14.2%)	4 (12.5%)	
Smoker			0.338
Non	45 (31.9%)	6 (18.8%)	
Current	57 (40.4%)	15 (46.9%)	
Ex	39 (27.7%)	11 (34.4%)	
Stage			>0.999
IIIB	35 (24.8%)	8 (25.0%)	
IV	106 (75.2%)	24 (75.0%)	
Histology			0.926
Adenocarcinoma	97 (68.8%)	22 (68.8%)	
Squamous cell cancer	40 (28.4%)	10 (31.2%)	
Unspecfied	4 (2.84%)	0 (0%)	
Overall response			0.447
PR	34 (24.1%)	11 (34.4%)	
SD	45 (31.9%)	10 (31.2%)	
PD	62 (44.0%)	11 (34.4%)	
Treated cycles			0.980
2	7 (5.0%)	1 (3.1%)	
3	31 (22.0%)	6 (18.8%)	
4	21 (14.9%)	6 (18.8%)	
5	10 (7.1%)	2 (6.3%)	
6	72 (51.1%)	17 (53.1%)	

Abbreviations: ANC, absolute neutrophil count; ALT, alanine transaminase level; AST, aspartate aminotransferase level; CLcr, estimated creatinine clearance; eGFR, estimated glomerular filtration rate; Non, none smoker; Ex, ex-smoker; PR, partial response; SD, stable disease; PD, progressive disease.

## Data Availability

Data supporting reported results are available from the corresponding author upon approval of a written request by Severance Hospital.

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
