# Peer review of "A Pharmacometric Model to Predict Chemotherapy-Induced Myelosuppression and Associated Risk Factors in Non-Small Cell Lung Cancer"

_pharmaceutics, 2022, doi:10.3390/pharmaceutics14050914_

Round 1
Reviewer 1 Report
The manuscript develops a K-PD model of absolute neutrophil count and identifies influencing factors that can be applied to the routine clinical setting of lung cancer chemotherapy. Data were collected retrospectively from electrical medical records for non-small cell lung cancer patients who underwent a combined chemotherapy regimen were used. Moreover, the developed model was used to predict the risk of severe neutropenia for individual patients.
In general, the structure and concise wording of the manuscript are adequate, which greatly facilitates its understanding by potential readers. On the other hand, tables and figures in the manuscript are adequate.
One limitation of the study is the low precision in the estimations of the inter-individual variabilities for some parameters. Authors are kindly requested to consider developing the model in NONMEM 7.5 incorporating lag differential equation solvers.
A limitation of the study is the poor precision in the estimations of interindividual variabilities for some parameters. Authors are kindly requested to consider developing the model into NONMEM 7.5 incorporating delay differential equations solvers (Delay differential equations based models in NONMEM. Yan et al. J Pharmacokinet Pharmacodyn. 2021 Dec;48(6):763-802. doi: 10.1007/s10928-021-09770-z.).
Reviewer 2 Report
Commentary on Pharmaceutics-1661077: Pharmacometric Model for Predicting Chemotherapy-Induced Myelosuppression and Associated Risk Factors in Non-Small Cell Lung Cancer by Kyemyung Park, Yukyung Kim, Mijeong Son, Dongwoo Chae, Kyungsoo Park of the Department of Pharmacology, Yonsei Yonsei University College of Medicine , Seoul, Korea.
The Authors presented quite interesting work on a new approach to predicting neutrophil changes using mathematical models. They developed a kinetic-pharmacodynamic model to predict the absolute neutrophil count during therapy in the non-small cell lung of cancer patients. The project may be useful in assessing the risk of fatal neutropenia, and this is its main value. It can also be used in the therapeutic monitoring of paclitaxel / cisplatin levels to avoid fatal neutropenia.
An interesting problem that emerged is the difference in susceptibility to chemotherapy-induced neutropenia between nondiabetic and diabetic subjects. Non-diabetic patients have been found to be more susceptible, but without a satisfactory explanation of the situation.
It is worth noting that the authors pointed to the limitations of their research. They focused on data from patients undergoing paclitaxel / cisplatin chemotherapy, while molecularly targeted therapies and immunotherapies are already used to treat NSCLC. Data were collected from only one clinical center, and relatively few data meant that some parameters were not presented in the study. The number of references in the discussion was limited to 6 research sources.
In general, the article is well organized, the data is legible, and presented to potential readers and pharmacometric specialists.
Explain the following test result.
- In the parameter IR50 there was a difference between
Qsex -0.334 and QDM + 0.485. This should be better explained.
- Could the K-PD model be useful for drugs that represent non-linear pharmacokinetics?
Reviewer 3 Report
The authors have completed a meaningful and highly scientific effort to explore and predict patient factors that contribute to reductions in ANC with combination chemotherapy in patients with advanced cancer. They have developed and R shiny tool that appears to be able to complete some bayesian estimation of patient risk, given input of some preliminary individual patient data points (online tool appears broken at present).
Comments:
The identifiability of the model parameters is not certain. Typically a bootstrap of the final model is performed to confirm identifiability and reproducibility of the final model parameters. Please perform the bootstrap evaluation with n = at least 200 replicates (prefer 1000) or complete an equivalent analysis to defend the estimability of the final model structure and parameters.
Tried the RShiny App and it gave an error. Set BSA, ANC, Sex, DM, Paclitaxel dose on Day 1, Cisplatin dose on Day 2, and the plot gave the result "An error has occurred. Check your logs or contact the app author for clarification." Please revise.
RShiny app: Please specify unit of simulation time (e.g. days) and unit of Y axis (e.g. 10^6 neutrophils per mL).
Recommendation: Have user be able to input weight/height and select method of calculating BSA to simplify use.
Please acknowledge that there would have been an opportunity to use popPK models for the chemotherapeutic drugs to simulate drug concentrations rather than using the mean approximation. This method would have helped to remove uncertainties in IIV in PD response associated with the influence of drug exposure on ANC.
